# Signal Enhancement of Selected Norepinephrine Metabolites Extracted from Artificial Urine Samples by Capillary Electrophoretic Separation

**DOI:** 10.3390/ijms252212227

**Published:** 2024-11-14

**Authors:** Piotr Kowalski, Natalia Hermann, Dagmara Kroll, Mariusz Belka, Tomasz Bączek, Ilona Olędzka

**Affiliations:** 1Department of Pharmaceutical Chemistry, Medical University of Gdańsk, 80-416 Gdańsk, Poland; piotr.kowalski@gumed.edu.pl (P.K.); natalia.hermann@gumed.edu.pl (N.H.); dagmara.kroll@gumed.edu.pl (D.K.); mariusz.belka@gumed.edu.pl (M.B.); tomasz.baczek@gumed.edu.pl (T.B.); 2Department of Nursing and Medical Rescue, Institute of Health Sciences, Pomeranian University in Słupsk, 76-200 Słupsk, Poland

**Keywords:** catecholamines, DHPG, 3D-printed dispersive solid-phase extraction, MHPG, micellar electrokinetic chromatography, liquid-liquid extraction, solid-phase extraction, VMA

## Abstract

The measurement of selected norepinephrine metabolites, such as 3,4-dihydroxyphenylglycol (DHPG), 3-methoxy-4-hydroxyphenylethylenglycol (MHPG), and vanillylmandelic acid (VMA), in biological matrices—including urine—is of great clinical importance for the diagnosis and monitoring of diseases. This fact has forced researchers to evaluate new analytical methodologies for their isolation and preconcentration from biological samples. In this study, the three most popular extraction techniques—liquid-liquid extraction (LLE), solid-phase extraction (SPE), and a new 3D-printed system for dispersive solid-phase extraction (3D-DSPE)—were investigated. Micellar electrokinetic chromatography (MEKC) with a diode array detector (DAD) at 200 nm wavelength was applied to the separation of analytes, allowing for the assessment of the extraction efficiency (R) and enrichment factor (EF) for the tested extraction types. The separation buffer (BGE) consisted of 5 mM sodium tetraborate decahydrate, 50 mM SDS, 15% (*v*/*v*) MeOH, 150 mM boric acid, and 1 mM of 1-hexyl-3-methylimidazolium chloride (the apparent pH of the BGE equaled 7.3). The EF for each extraction procedure was calculated with respect to standard mixtures of the analytes at the same concentration levels. The 3D-DSPE procedure, using DVB sorbent and acetone as the desorption solvent, proved to be the most effective approach for the simultaneous extraction and determination of the chosen compounds, achieving over 3-fold signal amplification for DHPG and MHPG and over 2-fold for VMA. Moreover, all extraction protocols used for the selected norepinephrine metabolites were estimated and discussed. It was also confirmed that the 3D-DSPE-MEKC approach could be considered an effective tool for sample pretreatment and separation of chosen endogenous analytes in urine samples.

## 1. Introduction

The compounds 3,4-Dihydroxyphenylglycol (DHPG), 3-methoxy-4-hydroxyphenylglycol (MHPG), and vanillylmandelic acid (VMA) are metabolic products of epinephrine (E) and norepinephrine (NE), which are part of the catecholamine (CA) group. These compounds are produced through the enzymatic actions of monoamine oxidase (MAO) and catechol-O-methyl transferase (COMT). After metabolism, they are released into the cerebrospinal fluid (CSF) and blood (specifically, plasma), and ultimately excreted from the body through urine. Their elevated levels in the blood are linked to stress and may result from excessive production of catecholamines by neuroendocrine tumor cells or be associated with certain mental health disorders [1,2].

DHPG is formed from NE by deamination to dihydroxyphenylglycolaldehyde (DOPEGAL) and reduction of the aldehyde by aldehyde reductase. Subsequently, DHPG is metabolized by COMT to form MHPG. An elevated NE:DHPG ratio in patients, could be the symptoms of inhibition activity of MAO or of aldehyde/aldose reductase [3]. Decreased DHPG levels are observed in Menkes disease [4], which affects infants and is a fatal neurodegenerative X chromosome-linked recessive disease. In turn, patients with depression resistant to dexamethasone exhibit increased excretion of DHPG [5]. The concentration levels of MHPG and DHPG in urine samples can be influenced by various factors, including age, gender, diet, health status, the use of medications and psychoactive substances, as well as the techniques used for sample collection and storage. Consequently, reference values for MHPG and DHPG in urine may vary depending on the laboratory and the testing methodology employed.

MHPG, a major metabolite of norepinephrine (NE), serves as a biomarker for evaluating nervous system function. It is used in the clinical diagnosis of various diseases, including Parkinson’s disease, vascular parkinsonism, depression, schizophrenia, mental disorders, bipolar disorder, and acute stroke [6,7,8,9,10,11,12,13]. The literature suggests that MHPG can serve as a marker in the diagnosis of pheochromocytomas or paragangliomas, as MHPG secretion from pheochromocytomas is significantly higher than from normal tissues [14,15]. MHPG is metabolized to VMA in the liver by the action of alcohol dehydrogenase. Measuring VMA in a 24 h urine sample is valuable for diagnosing pheochromocytoma and neuroblastoma [16]. VMA deficiency is linked to a congenital deficiency of DOPA decarboxylase, the enzyme responsible for converting DOPA to dopamine [17]. In patients with known hypertension, urinary VMA levels can be more than seven times higher, and in those with pheochromocytoma, the levels may exceed those in healthy individuals by over 35 times.

The determination of catecholamines (CAs) and their metabolites has evolved from early bioassays, colorimetric, and fluorometric methods to liquid chromatography-based techniques for measuring these compounds in plasma and urine [18]. Liquid chromatography with electrochemical detection (LC-ECD) or coupled with tandem mass spectrometry (LC-MS/MS) is among the most commonly used analytical techniques [19,20,21,22]. Immunoassays have also gained popularity for measuring plasma catecholamine metabolites; however, despite their high throughput, immunoassays suffer from limitations such as low sensitivity and specificity [23,24]. High-performance methodologies often require labor-intensive sample preparation and long chromatographic run times to minimize potential interferences that may co-elute with CAs. Consequently, there is still a need for rapid, efficient, and specific methods to isolate analytes from biological matrices. While most studies have focused on plasma samples, 24 h urine samples, blood, serum, and human and rat brain tissue have also been investigated [10,25,26,27]. Liquid-liquid extraction (LLE) has been widely used for isolating CA metabolites from urine. This method often involves the use of organic solvents to separate the analytes from the aqueous phase. For example, studies have reported using ethyl acetate or dichloromethane to extract, among others, MHPG, DHPG, and VMA from urine samples, achieving good recovery rates [28,29,30]. However, LLE may require large volumes of organic solvents and can be time-consuming. In a few studies, the sample preparation process before separation was based on the precipitation and elimination of proteins [25,26,30,31]. Solid-phase extraction (SPE) is another popular method for isolating these analytes. SPE utilizes a sorbent to capture the analytes from the liquid phase. Different types of sorbents, such hydrophilic-lipophilic balanced (HLB) [1,20,32], mixed strong anion exchange absorbent (MAX) [33], octadecyl (C18) [27], or alumina [34], have been employed to optimize extraction efficiency. Studies indicate that SPE offers improved selectivity and lower solvent consumption compared to LLE. For instance, a study demonstrated that using C18 columns with methanol as the eluent effectively isolated DHPG, MHPG, and VMA from urine samples [27]. Solid-phase microextraction (SPME) involves the use of a coated fiber that adsorbs analytes from the sample matrix. It offers several advantages, including solvent-free extraction, simplicity, and rapid analysis. Studies have shown that SPME can effectively extract catecholamine metabolites, including MHPG, DHPG, and VMA, from biological matrices. For instance, some studies reported high recovery rates for these analytes using SPME coupled with gas chromatography-mass spectrometry (GC-MS), demonstrating its potential for sensitive detection of catecholamine metabolites [35]. Recent advancements have introduced dispersive solid-phase extraction (DSPE), which combines the advantages of SPE and LLE. DSPE involves adding a sorbent to the sample matrix and mixing, allowing the analytes to adsorb onto the sorbent particles. This method has been shown to enhance extraction efficiency and reduce solvent usage.

In this study, we propose an alternative separation method using micellar-electrokinetic chromatography (MEKC), a high-resolution, environmentally friendly technique that enables rapid analysis while requiring minimal sample volume and solvents for the separation process. Since electromigration methods, such as MEKC, are generally less sensitive than HPLC techniques, it is essential to not only focus on the extraction efficiency of the analytes from the sample matrix but also on the potential for signal enhancement through the extraction process. Therefore, various approaches were tested in this study, which may prove promising and allow for the application of an optimal extraction method for the analysis of real samples.

The primary aim of this work was to assess the effectiveness of extraction techniques, including liquid-liquid extraction (LLE), solid-phase extraction (SPE) with different modifications, and dispersive solid-phase extraction (DSPE) utilizing custom 3D-printed components. The extraction methods were evaluated and compared for the analytes DHPG, MHPG, and VMA. Notably, the application of the newly designed 3D-printed DSPE with commercially available DVB sorbent for isolating these analytes has not been previously reported in the literature.

## 2. Results and Discussion

### 2.1. Optimization of Extraction Parameters

Even though various approaches have been reported for the determination of CAs in different biological samples, there is a lack of analytical solutions for the simultaneous determination of MHPG, DHPG and VMA using the MEKC-UV method. The goal of our study was to develop and optimize an extraction method for the simultaneous isolation and electrophoretic quantitation of selected norepinephrine pathway metabolites from urine samples.

In this study, artificial urine was used to evaluate the efficiency of various extraction methods, as its consistent composition facilitates method optimization. However, real urine is essential for final validation due to significant differences in matrix effects. Matrix effects, caused by co-eluting substances, can lead to ion suppression or enhancement in techniques like liquid chromatography-mass spectrometry (LC-MS), affecting analyte detection [36]. Real urine, a complex fluid with diverse endogenous compounds, presents more interfering substances than artificial urine, which typically lacks many metabolites and proteins. While matrix effects are crucial for method development in separation techniques, they are generally less significant during analyte isolation, where selective extraction is prioritized.

Although CAs have very similar structures, their chemical behaviors, namely interactions with various SPE sorbents, differ significantly. The catecholamines contain a catechol moiety, but differ with substituents and functional groups, which produces significant differences in hydrophobicity and pKa values (Table 1). The literature indicates that CAs are particularly sensitive to minor pH changes, and laboratory guidelines recommend adjusting urine samples to pH 3 to maintain their stability during storage [37]. However, the pH that best promotes the isolation of these analytes from the sample matrix may vary, depending on the pKa and log P values of the individual compounds. This parameter was therefore examined in our study to determine optimal conditions for analyte isolation. Some authors have indicated that the pH of urine samples was crucial for extraction efficiency of analytes and a wide pH variation (i.e., 4.5–8.0) of individual urine samples caused substantial pH differences for the pre-treated urine sample [25]. Therefore, in this study on different extraction approaches, we focused on adjusting the pH value of each sample to a narrow pH range (5.7–6.0).

In this study, three extraction approaches—LLE, SPE, and a new dispersive solid-phase extraction (DSPE) using a 3D-printed system—were tested for isolating MHPG, DHPG and VMA. Analyte signal intensity was used as a preliminary assessment of extraction efficiency. To identify the most effective conditions for analyte recovery during the development of an extraction method, analyte signal intensity can help compare different extraction procedures such as sorbent choice, solvent choice, pH, or extraction time. It is a useful solution, particularly during method development and optimization. However, we are aware that it is not a substitute for a full validation, which ensures the method’s accuracy, precision, selectivity, and robustness [36,38,39]. For each of the tested isolation techniques, both the extraction efficiency (R) and the potential for analyte signal enhancement (EF) were evaluated. The extraction efficiency was assessed according to Equation (1) (see Section 3.6) by conducting three independent measurements. If the extraction method yielded an extraction efficiency above 100%, this could result not only from the potential effectiveness of the approach in isolating the selected analytes from the matrix but also from the dissolution of the dry residue, after evaporating the eluent/desorbent, in 50 µL of sample buffer during the final stage of the procedure. Of particular interest were the procedures that allowed for signal enhancement, which was calculated using Equation (2) (see Section 3.6).

#### 2.1.1. Liquid-Liquid Extraction

In the initial stage, procedures based on liquid-liquid extraction with ethyl acetate as the organic phase were carried out. To select the most optimal extraction conditions, the influence of the matrix pH on extraction efficiency was evaluated. To acidify or alkalize the test sample, 0.1 M HCl or 25% ammonia solution was used. The best results were obtained when the pH of the sample matrix was between 5.5 and 5.8. This was probably due to the pKa values of the tested analytes, indicating that DHPG (−3, 9.21) and MHPG (−3, 9.91) are weak acids. For VMA, the pKa values (−4.1 and 3.11) indicate that it is a stronger acid (Table 1). Considering the log P values (−0.72 for DHPG, 0.11 for MHPG, and 0.43 for VMA), DHPG shows the most polar properties among the investigated compounds, while VMA is the least polar.

Based on the extraction efficiency of LLE procedures (see Table 2), it was determined that the optimal pH range for DHPG, VMA, and MHPG is between 5.5 and 5.8. For single LLE extractions, the best results were obtained at pH 4.6 for DHPG, pH 5.5 for MHPG, and pH 6.0 for VMA. The R values were 159.4% (±10.5), 125.2% (±5.9), and 197.8% (±5.7), respectively, which allowed for signal enhancement by 1.6 times for DHPG and 1.2 times for MHPG, and nearly a two-fold concentration increase for VMA. However, for double LLE, the highest R values for all analytes were obtained at pH 5.8. These results were more favorable than those from single LLE, enabling signal amplification by up to 2.2 times for DHPG and 1.6 times for VMA. In contrast, neutral and alkaline conditions proved unfavorable for MHPG extraction. A similar extraction method was used by Soda et al. for the analysis of plasma levels of HVA, free MHPG, total MHPG, and 5-HIAA in healthy individuals [30]. Ethyl acetate was applied for analyte extraction, with the optimal pH of 5.5 achieved by adding 1 mL of 0.5 M acetate buffer. However, it should be noted that LLE-based procedures are not highly efficient, do not allow for significant preconcentration of analytes, and require relatively large volumes of organic solvents. For these reasons, LLE is rarely used for the analysis of DHPG, MHPG, or VMA.

#### 2.1.2. Solid-Phase Extraction

In the next stage of the experiments, the SPE efficiency of analytes for seven types of sorbents was estimated (Table 3). The results, presented in Table 4, indicate that the recovery (R) and potential for signal enhancement for DHPG, MHPG, and VMA after isolation from urine samples using the SPE technique depend on the physicochemical properties of the analytes, the type of sorbents, and the eluting/desorbing agents.

For hydrophilic-lipophilic balanced (HLB) columns, where the sorbent is a hydrophilically modified styrene polymer, retention is primarily based on a reversed-phase mechanism. However, due to the hydrophilic modification, the sorbent is also prone to interactions with polar compounds. HLB sorbents offer good extraction recovery for non-polar to moderately polar acidic, neutral, and alkaline compounds, making them particularly suitable for the pretreatment of complex matrices such as blood, urine, and food. In our experiment with HLB columns, the highest recovery (R) was achieved using acetone as the eluent for DHPG (86.8 ± 2.8%) and VMA (83.4 ± 2.8%). For MHPG, the recovery was very low. Methanol (MeOH) proved to be a less effective eluent, with the best process efficiency obtained for the most hydrophobic compound, VMA, at 69.8 ± 6.3%. The ACN:MeOH mixture was ineffective as an eluent for HLB cartridges, providing an extraction efficiency of less than 37.6 ± 2.5%.

The octadecyl (C18) cartridges contain silica gel modified with octadecyl groups, enabling the extraction of non-polar compounds. In this case, the best extraction efficiency (R) values were achieved using methanol, which allowed enhancement factors (EF) of 3.0, 2.4, and 1.5 times for VMA, DHPG, and MHPG, respectively. The extraction results obtained by Yoshimura et al. also confirm high recovery for MHPG (81%) when C18 sorbent and a methanol-phosphate buffer mixture were used as the eluent [27]. Satisfactory results were also obtained with an eluting mixture of ACN:MeOH (50:50, *v*/*v*) for VMA and DHPG, where EF values were 1.9 times for VMA and 1.1 times for DHPG. Using acetone as the eluent, the best EF was achieved for VMA (1.5 times) and DHPG (1.2 times), while for MHPG, acetone was inefficient, yielding an extraction efficiency of 41.8 ± 2.9% without any signal enhancement. The better extraction efficiency for VMA is likely due to its lower hydrophilicity (log P 0.43) compared to DHPG (log P −0.72) or MHPG (log P 0.11).

Cyanopropyl (CN) sorbents, due to their strong and reversible interactions, allow for the extraction of polar compounds. Acetone was the only eluent used, providing a 1.4-fold signal enhancement for DHPG. However, unsatisfactory extraction efficiency was observed for MHPG and VMA, with values of 52.1 ± 4.0% and 67.8 ± 4.4%, respectively. The higher extraction efficiency and signal enhancement for DHPG can be attributed to its more polar properties.

Polymeric SPE for acidic, basic and neutral analytes (ABN) sorbents are a modified polymer phase of PS-DVB, consisting of a polystyrene divinylbenzene polymer for reversed-phase (hydrophobic) retention, with non-ionizing hydroxyl groups that ensure wettability without secondary interactions. ABN cartridges can be used to extract a wide range of acidic, basic, and neutral analytes from aqueous matrices, including biological fluids. According to the manufacturer, ABN sorbents are resistant to deconditioning, meaning they do not exhibit poor analyte–sorbent interactions due to phase collapse. One of the key advantages is the elimination of the need for conditioning and equilibration steps. Additionally, these sorbents offer protein exclusion, and their wash and elution protocols reduce or eliminate matrix interferences [40]. In our study, we followed the manufacturer’s recommended procedure, which did not require an activation step. After applying the urine sample, the sorbent was washed with a water:MeOH mixture (95:5, *v*/*v*), and pure MeOH was used as the eluent. However, under these conditions, unsatisfactory extraction efficiency was observed for DHPG and MHPG. For VMA, which has the most acidic properties among the analyzed compounds, the R value was only 21.2 ± 2.6%.

Anion exchange (AX) columns are filled with a sorbent modified with a quaternary amine group and a chloride counterion. This positively charged sorbent retains acidic (negatively charged) analytes through nonpolar and strong anion exchange interactions. According to the manufacturer’s recommended procedure, the sorbent was not activated, and the analytes were eluted using a mixture of MeOH and FA (98:2, *v*/*v*). After applying the sample, potential contaminants were removed with 1 mL of a 0.05 M ammonium acetate pH 6:MeOH mixture (95:5, *v*/*v*), followed by 1 mL of MeOH. However, this procedure was not effective for DHPG and MHPG, with a low R value for DHPG (9.1 ± 2.3%), and no effective for MHPG. The best results were obtained for VMA (53.4 ± 3.7%). This difference is likely due to the pKa values of the analytes. VMA, which has a carboxyl group, is retained on the AX sorbent. While DHPG also has a carboxyl group, its higher polarity compared to VMA may explain its weaker retention. MHPG, likely due to the absence of acidic groups, is not retained on the AX sorbent.

Cation exchange (CX) columns contain a sorbent modified with a sulfonic acid group, which retains basic (positively charged) analytes through nonpolar interactions and strong cation exchange functions. This sorbent is designed to extract positively charged basic/cationic compounds. The dual “mixed-mode” retention mechanism of these sorbents enables the use of 100% organic solvents during the wash step, effectively removing interferences without compromising analyte recovery. Analyte elution is achieved at high pH, neutralizing the positive charge on the analytes. The recommended eluent, a mixture of MeOH:NH_4_OH (95:5, *v*/*v*), was applied. Similar to AX sorbent, no preliminary activation of the CX sorbent was necessary, and after the sample was applied, the sorbent was washed with 1 mL of 0.05 M ammonium acetate (pH 6), followed by 1 mL of methanol. The highest extraction efficiency was observed for MHPG, with an R of 30.5 ± 3.9%, while for VMA, it was only 12.8 ± 3.5%. DHPG, being the most polar compound, was not effectively extracted under these conditions.

SPE columns with polystyrene–divinylbenzene (PS–DVB) sorbent are commonly used for retaining neutral and aromatic compounds, making them suitable for screening applications involving a broad range of analytes. In the case of SPE with PS–DVB cartridges, three types of eluents were tested: pure acetone, an ACN:MeOH mixture (50:50, *v*/*v*), and a CH_2_Cl_2_:IPA:NH_4_OH mixture (78:20:2, *v*/*v*/*v*). As shown in Table 4, the highest extraction efficiency for DHPG (140.4 ± 3.5%) was achieved with acetone as the eluent, leading to a signal enhancement of up to 1.4 times. In contrast, the extraction efficiencies for MHPG and VMA were 50.5 ± 2.9% and 67.5 ± 4.1%, respectively, with no potential for signal enhancement. The other eluents tested were ineffective for tested analytes.

#### 2.1.3. Dispersive Solid-Phase Extraction with a 3D-Printed System

Previously obtained results for LLE and SPE procedures were compared with those obtained using the DSPE technique, incorporating a new 3D-printed system. A commercially available Dow Styrene DVB sorbent was placed in the printed system. This sorbent can effectively extract a wide range of analytes from polar samples. By adjusting the sample pH, as well as the wash and elution solvents, DVB sorbents can be used for the analysis of acidic, basic, and neutral (both polar and non-polar) compounds. In this experiment, 30 mg of sorbent beads were weighed and placed in the extraction chamber of the 3D-printed system to facilitate the adsorption of analytes. The extraction process followed the procedure described in our previous publication, where it was tested for imipramine and carbamazepine quantification [41].

In this paper, we present a new application of our previously described 3D-printed vessels for the isolation of DHPG, MHPG, and VMA from urine samples. We demonstrated that the dispersive extraction of DHPG, MHPG, and VMA with the aid of a Styrene-DVB sorbent is both effective and convenient. A schematic diagram of the 3D-DSPE-based extraction procedure is shown in Figure 1. Moreover, an evaluation of the extraction efficiency of the sorbent, potential for signal enhancement, and repeatability was conducted.

The results obtained for both desorbents used in this experiment (acetone and the ACN:MeOH mixture (50:50, *v*/*v*)) for samples at pH 5.8, turned out to be very promising. Satisfactory yields of the extraction process were obtained with the possibility of signal amplification. Acetone made it possible to achieve over 3-fold signal amplification for DHPG and MHPG and over 2-fold for VMA. Only slightly lower results were provided by the ACN:MeOH mixture (50:50, *v*/*v*) and were 2.9 times for DHPG, 2.4 times for MHPG, and 2.1 times for VMA, respectively (Table 5). More favorable results for the same type of sorbent and desorbent used in SPE procedure, may be attributed to a looser distribution of sorbent beads in the DSPE procedure, which increases the contact surface area with the sample matrix and the analytes contained within. In our 3D-DSPE procedure, both the extraction and desorption processes were enhanced by mechanically mixing the sample on a magnetic stirrer.

Generally, the DSPE technique is considered advantageous for sample preparation due to its simplicity and high recovery rates. DSPE is viewed as a variant of SPE, where a powdered sorbent is added directly to the liquid sample. Continuous stirring allows the analyte to adsorb onto the surface of the sorbent particles. The liquid phase is then separated from the solid sorbent, and an appropriate desorption solvent is added. One of the main challenges is the difficulty in effectively separating the powdered sorbent from the liquid extract. It is worth noting that DSPE has the significant advantage of requiring a much smaller amount of solvent. For this reason, novel solutions in dispersive extraction are still being explored to improve the separation of the sorbent from the sample matrix and achieve more efficient extraction results.

### 2.2. Optimization of Micellar-Electrokinetic Chromatography as Separation Method

The separation parameters of the MEKC-UV (DAD) method were developed, optimized, and discussed in our previous paper [29]. In that publication, we provided a detailed analysis of the optimization of the separation buffer composition. Due to the diverse pKa values of the selected panel of CAs, their simultaneous determination is an analytical challenge. The chemical structures of these compounds contain multiple functional groups that can ionize over a wide pH range, making the selection of optimal buffer ingredients a complex task. In our study, a borate buffer was chosen, as catechol compounds and some substituted catechol (like CAs) can become charged in a weakly alkaline electrolyte. Once ionized, the hydroxyl groups are capable of forming complexes with borate ions, allowing for adequate electrophoretic mobility. However, due to insufficient separation of all analytes, the addition of surfactants and an organic modifier was explored. It was shown that the presence of 50 mM SDS and 15% methanol in the separation buffer significantly improved separation efficiency. A borate buffer without SDS or with SDS concentrations below 50 mM did not provide satisfactory separation of the compounds. Additionally, the 15% (*v*/*v*) methanol as an organic modifier improved separation efficiency by influencing buffer viscosity.

The optimized separation buffer, composed of 5 mM sodium tetraborate decahydrate, 50 mM SDS, 15% MeOH (*v*/*v*), and 150 mM boric acid, was further enhanced with the addition of an imidazolium-based ionic liquid (1 mM 1-hexyl-3-methylimidazolium chloride). The application of this ionic liquid in the background electrolyte (BGE) created a dynamic double-coating effect, resulting in greater repeatability of analyte migration times and satisfactory separation of the analytes [42]. A representative electropherogram showing the separation of DHPG, VMA, and MHPG is presented in Figure 2.

It should be pointed out that electromigration-based procedures require capillary rinsing after each run to ensure proper repeatability of migration times in individual analyses. Regarding the potential carry-over effect, baseline control was also performed by analyzing the separation buffer alone. To check for any residual analytes carried over from one sample to the next, a pure separation buffer was performed after the analysis of a sample containing high concentrations of the analytes. No analyte signals were detected, indicating that there was no carry-over from the previous sample.

## 3. Materials and Methods

### 3.1. Chemicals and Reagents

The analytes 3,4-dihydroxyphenylglykol (DHPG), 3-methoxy-4-hydroxyphenylglycol (MHPG), and vanililmandelic acid (VMA) and the reagents acetone, acetonitrile (ACN), chlorophorm, dichloromethane (DCM), boric acid, formic acid (FA), sodium tetraborate decahydrate, sodium hydroxide (NaOH), sodium hydroxycarbonate, sodium dodecyl sulphate (SDS), and 1-hexyl-3-methylimidazolium chloride were purchased from Merck (Darmstadt, Germany). The reagents ammonia, ammonium acetate, diethyl ether, and ethanol (96%) were obtained from POCH (Gliwice, Poland). Methanol (MeOH), n-hexan, and propan-2-ol (IPA) were supplied by VWR International Poland (Gdańsk, Poland). All chemicals were of analytical grade and were applied without further purification. The Capillary Regenerator Basic Wash (0.1 M sodium hydroxide) was purchased from Beckman Coulter (Brea, CA, USA). Different types of SPE cartridges such as: Evolute^®^ Express ABN, 30 mg/1 mL; Evolute^®^ Express AX 30 mg/1 mL; and Evolute^®^ Express CX, 30 mg/1 mL were purchased from Biotage (Uppsala, Sweden); cartridges LiChrolut^®^ C18 and CN were purchased from Merck (Darmstadt, Germany); DVB cartridge 30 mg/1 mL was supplied by UCT (Bristol, PA, USA); and the Supel™-Select HLB, 30 mg/1 mL and the Styren-DVB (Styre screen) sorbent were purchased from Supelco/Sigma-Aldrich (St. Louis, MO, USA). The deionized (DI) water (18.2 MΩ cm) used in all experiments was obtained from Milli-Q apparatus from MiliporeSigma (Burlington, MA, USA). The 3D-printed DSPE system consisting of four printed elements was fabricated in Department of Pharmaceutical Chemistry, Medical University of Gdańsk, Poland. Artificial urine (SAE0074-50ML) was purchased from Sigma-Aldrich (St. Louis, MO, USA).

### 3.2. Analytical Equipment

MEKC separations were performed on a P/ACE MDQ Capillary Electrophoresis System (Beckman Coulter, Fullerton, CA, USA) equipped with an automatic sample dispenser and a UV/VIS-DAD detector set to an analytical wavelength of 200 nm. The MDQ system also included a capillary thermostat to maintain a constant temperature throughout the separation process. Data acquisition was managed using 32 Karat 8.0 software (Beckman, Fullerton, CA, USA). Sample preparation was conducted using an SPE system (Agilent Vac Elut SPS 24 Manifold, Santa Clara, CA, USA), and sample evaporation was carried out with a CentriVap vacuum concentrator (Labconco, Kansas City, MO, USA). Buffer and solution pH values were measured with a pH meter (Mettler Toledo, Warsaw, Poland).

### 3.3. Preparation of Stock and Working Solutions

Stock standard solutions of DHPG, VMA, and MHPG were prepared by accurately weighing 1.0 mg of each analyte and dissolving it in 1 mL of methanol (MeOH). These solutions were stored in closed containers in a freezer at −20 °C for up to one month. To ensure analyte stability, working standard solutions were freshly prepared each day by diluting the stock solutions in water and storing them at 4 °C in closed containers for a maximum of 8–9 h. Artificial human urine samples were spiked with the analytes to a final concentration of 10 μg/mL. The samples were then either analyzed directly or processed according to the investigated extraction procedures. After the eluent/desorbent was evaporated, the dry residue was re-dissolved in 50 μL of sample buffer (2 mM sodium tetraborate decahydrate) and separated using the MEKC method.

### 3.4. MEKC Conditions

MEKC separations were conducted using a PACE/MDQ system equipped with a DAD-UV detector set to a wavelength of 200 nm. An uncoated fused-silica capillary (75 µm i.d./365 µm o.d., 50 cm effective length, 60.2 cm total length) was maintained at 22 ± 0.1 °C. The analysis time was 8 min, with hydrodynamic injection performed at 0.5 psi (3.45 kPa) for 30 s. A voltage of 25 kV was applied, producing a current of approximately 50 µA. The background electrolyte (BGE) consisted of 5 mM sodium tetraborate decahydrate, 50 mM SDS, 15% (*v*/*v*) methanol, 150 mM boric acid, and 1 mM 1-hexyl-3-methylimidazolium chloride, with an apparent pH of 7.3. The sample buffer used was 2 mM sodium tetraborate.

Capillary conditioning was carried out at 20 psi (137.9 kPa) with the following steps: an initial rinse with 0.1 M NaOH for 5 min, followed by a rinse with deionized Milli-Q water for 10 min, and a final rinse with BGE for 5 min at 20 psi. Each day before analysis, the capillary was washed with 0.1 M NaOH for 1 min, followed by deionized Milli-Q water and BGE for 5 min. Between runs, the capillary was rinsed with 0.1 M NaOH for 1 min, deionized water, and BGE for 1 min. At the end of each day, the capillary was rinsed with 0.1 M NaOH for 2 min and deionized Milli-Q water for 5 min.

### 3.5. Extraction Procedures

#### 3.5.1. Liquid-Liquid Extraction Procedure (LLE)

The LLE procedures were performed using 1 mL of artificial urine samples, adjusted to the appropriate pH and spiked with a working solution of the analytes to achieve a final concentration of 10 µg/mL. The samples were mechanically shaken for 2 min, followed by the addition of 1.5 mL of ethyl acetate as the extraction solvent, and shaken again for 5 min. The samples were then centrifuged at 4000× *g* for 4 min. The upper organic phase was carefully separated and transferred to a clean Eppendorf tube. In the case of double LLE, another 1.5 mL portion of ethyl acetate was added to the aqueous phase, and the extraction process was repeated. The organic phases from both extractions were combined and evaporated to dryness at 45 °C (Labconco, Kansas City, MO, USA). The resulting residue was dissolved in 50 μL of 2 mM sodium tetraborate and analyzed using the developed MEKC method.

#### 3.5.2. Solid-Phase Extraction Procedure (SPE)

SPE experiments were carried out on hydrophilic-lipophilic balanced (HLB), octadecyl (C18), cyanopropyl (CN), anion exchange (AX), cation exchange (CX), ABN and DVB cartridges. The sorbents HLB, C18, and CN were previously activated with 1 mL of MeOH and 1 mL of DI water. The sorbents AX, CX, ABN, and DVB, in accordance with the manufacturer’s instructions, do not require prior activation. The 1 mL of artificial human urine, adjusted to appropriate the pH value and spiked with the analytes at a concentration of 10 μg/mL, was loaded into cartridges. Next, the washing step was applied using the solution presented in Table 3, followed by drying in a vacuum for 5 min. The analytes were eluted from the SPE cartridges with 1 mL of every one of the tested eluents shown in Table 3. Then, the eluent was evaporated to dryness at 45 °C in vacuum (Labconco, Kansas City, MO, USA) for 1 h and the residue was dissolved in 50 μL of sample buffer and separated using the MEKC method.

#### 3.5.3. Dispersive Solid-Phase Extraction Procedure (DSPE)

DSPE was realized based on our earlier publication concerning a newly developed, fully 3D-printed device [41]. To activate the sorbent, the extraction procedure was preceded by the conditioning of the styrene DVB sorbent, which was performed in the 3D-printed vessel in the following manner: 1 min of shaking the styrene DVB sorbent in methanol, centrifugation, and then 2 min of milder shaking in deionized water followed by next centrifugation. The extraction with 30 mg of styrene DVB included two stages: sorption and desorption. For sorption, 1 mL of artificial urine sample, adjusted to pH 5.8 and with analytes at concentration 10 µg/mL, was applied. For desorption, 1 mL of acetone or ACN:MeOH mixture (50:50, *v*/*v*) was used. Both steps lasted 10 min, and the probes were stirred constantly in a magnetic stirrer (at 1000 rpm). The desorption phase was evaporated to dryness in a vacuum (Labconco, Kansas City, MO, USA) for 1 h. The resulting residue was dissolved in 50 μL of sample buffer and analyzed using the elaborated MEKC method.

### 3.6. Data Analysis

The extraction results for analytes from artificial urine samples were compared with those obtained from a standard solution containing each analyte at a concentration of 10 µg/mL. After the extraction process, two key parameters—extraction efficiency (R) and enrichment factor (EF)—were calculated for each analyte. Since the extraction methods were still in the optimization phase and had not yet been validated, the analytical instrument’s response, measured as peak height, was used to assess these parameters.

Extraction efficiency (R) was determined by comparing the signal intensity (peak height, H) of each analyte after extraction using the MEKC method to the signal intensity (peak height, H_0_) of the analyte in an untreated sample of the same concentration (10 µg/mL). This ratio provided a measure of the effectiveness of each extraction method. Equation (1). Extraction efficiency (R):R = H/H_0_ × 100%(1)
where: R—extraction efficiency; H—signal height of the analyte after extraction, sample evaporation and dissolution in 2 mM borax and MEKC analysis; H_0_—signal height of an analyte at the same concentration level after MEKC analysis without an extraction procedure.

While analyte recovery is commonly used to evaluate the efficiency of extraction techniques, it cannot be calculated when the method has not been validated and the post-extraction analyte concentration is unknown [36,38,39]. In such cases, the apparatus response—expressed as peak height or area—is used for signal analysis. In this study, peak heights, calculated as the average of three measurements for each analyte across each extraction procedures, were used in the calculations (Equation (1)).

The enrichment factor (EF) (Equation (2)) was calculated based on the understanding that the analytes were not directly measured during the extraction phase but were instead dissolved in only 50 µL of sample buffer after evaporation (offline preconcentration of analytes). Equation (2). Enrichment factor (EF):EF = H/H_0_(2)
where: EF—Enrichment factor; H and H_0_—as described above.

## 4. Conclusions

Effective methods for isolating analytes from biological matrices should facilitate the determination of low concentrations of endogenous substances in biological samples. The extremely low concentrations of catecholamines and their metabolites, including DHPG, MHPG, and VMA, along with their diverse physicochemical properties, necessitate a careful selection of extraction procedures and parameters. This study presents the first application of a new 3D-printed DSPE system for extracting compounds from the norepinephrine metabolic pathway (DHPG, MHPG, and HVA) from biological samples, comparing it with two different extraction methods based on LLE and SPE. Given the low concentrations of analytes in the sample matrix, we also focused on the potential for signal amplification, expressed as the enhancement factor, achieved during sample preparation. Among the LLE techniques, the greatest signal amplification was achieved through a modified double process at a sample pH of 5.8. The most efficient procedures for all analytes utilized C18 columns with methanol as the eluent. The results obtained from the new 3D-printed DSPE system with Styrene-DVB sorbent and acetone as the desorbent confirmed the significant potential of this solution for effective extraction and signal amplification for all tested analytes, reaching up to a 3-fold increase. Additionally, the developed MEKC method, supported by the proposed 3D-printed DSPE system with Styrene-DVB sorbent, can serve as an offline preconcentration technique for the simultaneous isolation and determination of these catechol compounds in urine samples for diagnostic purposes.

## Figures and Tables

**Figure 1 ijms-25-12227-f001:**
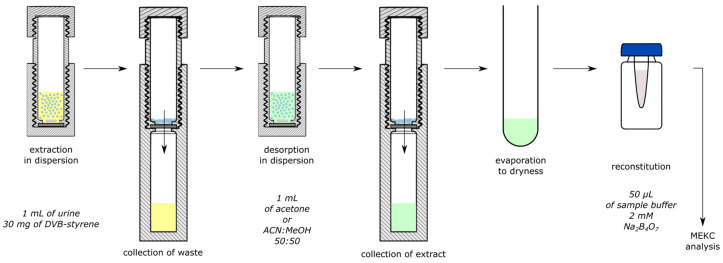
Scheme of the general 3D-DSPE procedure for the extraction and desorption of the selected NE metabolites by Dow Styrene DVB sorbent from artificial urine samples.

**Figure 2 ijms-25-12227-f002:**
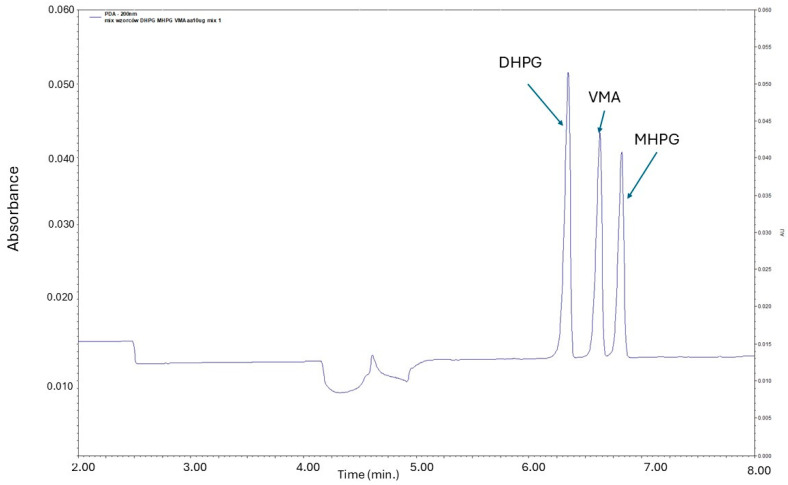
Electropherogram of standard solution of DHPG, VMA, and MHPG at a concentration of 10 µg/mL. Separation conditions: applied voltage 25 kV; capillary total length 50.2 cm and 75 µm i.d.; λ = 200 nm; injection time 30 s (at 0.5 psi); analysis time 8 min. The BGE consisted of 5 mM sodium tetraborate decahydrate, 50 mM SDS, 15% (*v*/*v*) MeOH, 150 mM boric acid, and 1 mM of 1-hexyl-3-methylimidazolium chloride (the apparent pH of the BGE equaled 7.3).

**Table 1 ijms-25-12227-t001:** The analytes and their physicochemical parameters.

Analyte	Chemical Structure	pKa Values	Log P (Experimental)	Water Solubility [g/L]
**Zwitterion compounds**				
DHPG	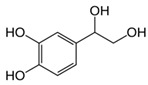	−3 9.21	−0.72	16.7
MHPG	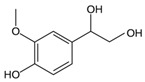	−3 9.91	0.11	8.99
**Compound with the acidic nature**				
VMA	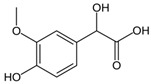	−4.1 3.11	0.43 0.94	5.16

**Table 2 ijms-25-12227-t002:** The results of average extraction efficiency (R) with SD value and enhancement factor (EF) for LLE based extraction procedures depending on pH of the sample matrix (n = 3).

LLE Based Extraction Procedures		DHPG		MHPG		VMA	
		R ± SD (%)	EF	R ± SD (%)	EF	R ± SD (%)	EF
single process	pH 4.6	159.4 ± 10.5	1.6	11.6 ± 1.3	-	103.8 ± 5.5	1.0
	pH 5.5	*no*	-	125.2 ± 5.9	1.2	103.5 ± 6.3	1.0
	pH 6.0	140.7 ± 4.9	1.4	*no*	-	197.8 ± 5.7	1.9
	pH 9.2	151.4 ± 6.0	1.5	*no*	-	18.6 ± 3.3	-
double process	pH 5.8	218.2 ± 4.2	2.2	192.2 ± 8.0	1.9	157.7 ± 8.8	1.6
	pH 7.4	209.9 ± 5.5	2.1	*no*	-	166.7 ± 4.2	1.6

*no*—no response signal for this procedure.

**Table 3 ijms-25-12227-t003:** The SPE extraction conditions depending on the type of the sorbent.

Type of Sorbent	Activation Step	Washing Step	Elution Step
HLB	1 mL of MeOH followed by 1 mL of DI water	1 mL of mixture water:MeOH (9:1, *v*/*v*)	1 mL of ACN:MeOH (50:50, *v*/*v*) or 1 mL of acetone or 1 mL of MeOH
C18			
CN			1 mL of acetone
AX	-	1 mL of 0.05 M ammonium acetate pH6:MeOH (95:5, *v*/*v*) followed by 1 mL of MeOH	1 mL of MeOH:formic acid (98:2, *v*/*v*)
CX	-		1 mL MeOH:ammonium hydroxide 25% (95:5, *v*/*v*)
Evolute ABN	-	1 mL of water:MeOH (95:5, *v*/*v*)	1 mL of MeOH
Styren DVB	-	1 mL of DI water	1 mL of CH_2_Cl_2_:IPA:NH_4_OH (78:20:2, *v*/*v*/*v*)

**Table 4 ijms-25-12227-t004:** The results of extraction efficiency (R) with SD value and enhancement factor (EF) for SPE-based extraction procedures depending on sorbent and eluent (n = 3).

SPE-Based Extraction Procedures		DHPG		MHPG		VMA	
		R ± SD (%)	EF	R ± SD (%)	EF	R ± SD (%)	EF
Sorbent	Eluent						
HLB	MeOH	59.5 ± 2.7	-	*no*	-	69.8 ± 6.3	-
	Aceton	86.8 ± 2.8	-	*no*	-	83.4 ± 2.8	-
	ACN:MeOH 50:50	18.2 ± 1.4	-	*no*	-	37.6 ± 2.5	-
C18	MeOH	239.9 ± 8.6	2.4	153.2 ± 3.9	1.5	300.5 ± 12.7	3.0
	Aceton	120.4 ± 4.8	1.2	41.8 ± 2.9	-	152.4 ± 3.4	1.5
	ACN:MeOH 50:50	111.3 ± 6.3	1.1	62.2 ± 2.7	-	190.8 ± 9.5	1.9
CN	Aceton	121.6 ± 5.7	1.2	52.1 ± 4.0	-	67.8 ± 4.4	-
ABN	MeOH	11.1 ± 1.3	-	*no*	-	21.2 ± 2.6	-
AX	MeOH:FA 98:2	9.1 ± 2.3	-	*no*	-	53.4 ± 3.7	-
CX	MeOH:NH_4_OH 25% 95:5	*no*	-	30.5 ± 3.9	-	12.8 ± 3.5	-
Styre DVB	Aceton	140.4 ± 3.5	1.4	50.5 ± 2.9	-	67.5 ± 4.1	-
	ACN:MeOH 50:50	*no*	-	*no*	-	*no*	-
	DCM:IPA:NH_4_OH	*no*	-	*no*	-	*no*	-

*no*—no response signal for this procedure.

**Table 5 ijms-25-12227-t005:** The results of extraction efficiency (R) with SD value and enhancement factor (EF) for 3D-DSPE-based extraction procedures depending on eluent (n = 3).

3D-DSPE-Based Extraction Procedures		DHPG		MHPG		VMA	
		R ± SD (%)	EF	R ± SD (%)	EF	R ± SD (%)	EF
Sorbent	Eluent						
DVB 30 mg	Aceton	350.6 ± 16.5	3.5	324.4 ± 7.1	3.2	228.9 ± 11.0	2.3
	ACN:MeOH 50:50	292.2 ± 15.4	2.9	249.3 ± 11.2	2.5	214.9 ± 10.5	2.1

## Data Availability

Data is contained within the article.

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
