# Peer review of "Signal Enhancement of Selected Norepinephrine Metabolites Extracted from Artificial Urine Samples by Capillary Electrophoretic Separation"

_ijms, 2024, doi:10.3390/ijms252212227_

Round 1
Reviewer 1 Report
Comments and Suggestions for Authors
This manuscript has serious flaws and, in my opinion, it cannot be accepted for publication in IJMS. Below are summarized a few concerns I have raised.
-Line 105. ...suffer should read ...suffers...
RESULTS: Par.2.1 .This paragraph (lines 122-171) presents results obtained by other authors and previously published. In my opinion it has nothing to do with the current work. Rather, I would see this whole paragraph as part of the Introduction section. In this context, the current Introduction must be widely shortened and totally re-written.
-Par.2.2 . As underlined above, the tendency of the authors is to dwell with the description of general concepts instead of focusing on their experimental results. May depend on the fact that these results are missing ? Again, lines 190-216 are not results and must not stay there !
-Table 4 is a mass. I cannot understand these data.
-Par.2.2.3. I would not repeat myself but the same problem as above arises in this paragraph. Lines 316-336 are completely inconsistent as results.
Provided that the method described is efficient on synthetic urine. Did the authors ever apply it on real urine ? Which are the results they have obtained ?
Author Response
Reviewer(s)' Comments to Author:
Reviewer 1
This manuscript has serious flaws and, in my opinion, it cannot be accepted for publication in IJMS. Below are summarized a few concerns I have raised.
Answer: We sincerely thank the reviewer for their valuable feedback and thoughtful evaluation of our manuscript. In response to the reviewer's comments, we have completely revised the manuscript, ensuring a clearer focus on the primary objective of our study: assessing the effectiveness of different extraction techniques for DHPG, MHPG, and VMA. These techniques include liquid-liquid extraction (LLE), solid-phase extraction (SPE) with various modifications, and dispersive solid-phase extraction (DSPE) using custom-designed 3D-printed components, which was first used for extraction procedure for selected catecholamines. A notable contribution of this work is the application of the newly designed 3D-printed DSPE combined with commercially available DVB sorbent, which, to our knowledge, has not been previously reported for isolating these analytes.
Additionally, we propose an alternative separation method using micellar-electrokinetic chromatography (MEKC), a high-resolution and environmentally friendly technique that allows rapid analysis with minimal sample volume and solvent usage. Recognizing that electromigration methods such as MEKC are typically less sensitive compared to HPLC techniques using spectrophotometric detection, we placed particular emphasis on not only optimizing extraction efficiency but also exploring the potential for signal enhancement during the extraction process. Various approaches were tested in this study, offering promising results that may facilitate the application of an optimal extraction method for analyzing real samples.
-Line 105. ...suffer should read ...suffers...
Answer: It was corrected.
RESULTS: Par.2.1 .This paragraph (lines 122-171) presents results obtained by other authors and previously published. In my opinion it has nothing to do with the current work. Rather, I would see this whole paragraph as part of the Introduction section. In this context, the current Introduction must be widely shortened and totally re-written.
Answer: Thank you for this comment. Based on the reviewer's suggestion, the paragraph in question has been significantly shortened, with key information relocated to the introduction section. Introduction has been totally re-written.
-Par.2.2 . As underlined above, the tendency of the authors is to dwell with the description of general concepts instead of focusing on their experimental results. May depend on the fact that these results are missing ? Again, lines 190-216 are not results and must not stay there !
Answer: It was completely corrected in current version of the manuscript. The results have been presented more clearly.
-Table 4 is a mass. I cannot understand these data.
Answer: The table of results has been revised and divided into sections corresponding to each evaluated extraction technique for improved clarity. Due to the fact that the previous version provided only average values, in the revised version the results of extraction efficiency have also been supplemented with standard deviation values, as each approach was repeated at least three times. Regarding the potential for analyte signal amplification, enrichment factor (EF) values are reported only when the extraction efficiency exceeded 100%, achieved through both effective isolation conditions and the final dissolution of the sample in just 50 microliters of buffer. This criterion ensured that signal amplification was present. We hope that this revised data presentation is more appropriate. Additionally, we have included explanations in the manuscript text about the use of analyte signal intensity in the optimization of the extraction method, citing a well-recognized source for support [1,2].
[1] https://www.fda.gov/files/drugs/published/Bioanalytical-Method-Validation-Guidance-for-Industry.pdf
[2] "Principles and Practice of Bioanalysis" (2nd Edition) by Richard F. Venn: Chapters on analytical method validation, which include discussions of the use of the analyte intensity signal during the initial stages of method development through to full validation procedures.
-Par.2.2.3. I would not repeat myself but the same problem as above arises in this paragraph. Lines 316-336 are completely inconsistent as results.
Answer: The content of this paragraph (2.1.3 in current version of the manuscript) has been corrected.
Provided that the method described is efficient on synthetic urine. Did the authors ever apply it on real urine ? Which are the results they have obtained ?
Answer: Artificial urine and real urine can differ in terms of matrix effects, especially when it comes to the detection of target compounds such as drugs or metabolites. Matrix effects refer to the influence of co-eluting substances from the sample matrix on the analytical detection of target analytes. Real urine is a complex biological fluid containing a wide variety of endogenous compounds such as proteins, salts, urea, creatinine, metabolites, and organic compounds whereas artificial urine is typically designed to mimic the main components of real urine (such as urea, creatinine, and electrolytes). Artificial urine is less complex and typically contains fewer interfering substances.
However, artificial urine is useful for method development and validation, but it may not perfectly replicate the real matrix effects observed with natural urine, particularly for complex drug metabolites. Artificial urine is often used for method development because it provides a consistent, reproducible matrix that can be used to optimize detection conditions.
However, we are aware that for final method validation, it is important to test with real urine samples to accurately assess matrix effects under actual analytical conditions.
We are aware also that confirming the applicability of the proposed solutions requires full validation and the use of real urine samples. This is the focus of our efforts in the near future.
Reviewer 2 Report
Comments and Suggestions for Authors
The present work demonstrates the use of MEKC for the determination of three catecholamine metabolites. It also provides a more comprehensive study of preconcentration techniques, including the relatively new DSPE technique.
The work is certainly original and brings some new insights, but one cannot help but notice that much of the work is a routine application of the authors' previous study. It differs only in its focus on different analytes.
I have the following comments on the article:
1. Correct the name of DHPG in the abstract (line 9) and I chemicals (line 369). It is written partially in Polish.
2. The sentence on lines 33-35 must be rewritten. It doesn't quite make sense in its present form.
3. The introduction to the article is very comprehensive in terms of the physiology of the substances studied and their pathways in the body. I recommend shortening and, on the contrary, strengthening the chemical level of the problem.
4. The paragraph from line 99 onwards lacks citations to the colorimetric and fluorimetric methods for the determination of CAs mentioned by the authors.
5. Line 171 - Did the authors try to increase the sensitivity of the determination by any of the mentioned techniques directly in the separation capillary (some kind of sample stacking etc)?
6. Figures 2 and 3 are of poor quality and need to be redone. In particular, the axis descriptions are not legible.
7. Table 2 - What does the acronym DHPG stand for? The structure looks like dihydroxyphenylglycine, but in the text, you talk about 3,4-dihydroxyphenylglycol. In addition, the table does not cite the source from which the pKA values were taken. For the structures reported, I find it very unlikely that any group would be acidic enough to have a negative pKA.
8. When discussing the choice of pH for LLE, pH values are given to two decimal places. How has pH been measured to this extreme precision? Do you think it is relevant to consider hundredths of a unit of pH if the pH is roughly 3 units different from pKAs?
Author Response
Reviewer 3
The present work demonstrates the use of MEKC for the determination of three catecholamine metabolites. It also provides a more comprehensive study of preconcentration techniques, including the relatively new DSPE technique. The work is certainly original and brings some new insights, but one cannot help but notice that much of the work is a routine application of the authors' previous study. It differs only in its focus on different analytes.
We appreciate the reviewer's interest in the manuscript and their valuable suggestions for improving its quality. In response to these suggestions, we have made efforts to clarify any doubts and enhance the manuscript. It has been completely rewritten, taking into account all of the reviewer's recommendations. We kindly ask that you re-evaluate the revised version.
I have the following comments on the article:
- Correct the name of DHPG in the abstract (line 9) and I chemicals (line 369). It is written partially in Polish.
Answer: Thank you for this comment. It was corrected.
- The sentence on lines 33-35 must be rewritten. It doesn't quite make sense in its present form.
Answer: Thank you for this comment. According to suggestion of other reviewer this sentence was removed from current version of the manuscript.
- The introduction to the article is very comprehensive in terms of the physiology of the substances studied and their pathways in the body. I recommend shortening and, on the contrary, strengthening the chemical level of the problem.
Answer: Thank you for this comment. Similar comments were also made by other reviewers. The introduction has been completely rewritten in accordance with the suggestions. The descriptions of the analytes studied have been shortened, while the sections describing the analytical techniques used for the isolation of analytes from biological matrices have been expanded, and attention has been paid to highlighting their advantages and disadvantages.
- The paragraph from line 99 onwards lacks citations to the colorimetric and fluorimetric methods for the determination of CAs mentioned by the authors.
Answer: Based on the literature data and Figure contained in the publication G. Eisenhofer, M. Peitzsch, Laboratory Evaluation of Pheochromocytoma and Paraganglioma Clinical Chemistry – Review; 60:12, 1486–1499 (2014), we established citations to the colorimetric and fluorimetric methods for the determination of CAs.
- Line 171 - Did the authors try to increase the sensitivity of the determination by any of the mentioned techniques directly in the separation capillary (some kind of sample stacking etc)?
Answer: We applied one of the basic on-line signal enhancement techniques in the capillary, known as FASS (field-amplified sample stacking). In this approach, the conductivity of the sample buffer must be lower than the conductivity of the separation buffer. As a result, after injecting the sample into the capillary, filled with separation buffer, and applying voltage, the analytes rapidly accelerate within the sample zone and stack at the boundary between the two buffers. This results in signal enhancement and improves the method's detection limit. However, it is still inferior to chromatographic techniques due to the short optical path, which is limited to the cross-section of the capillary. Therefore, there is significant potential in combining separation solutions in electromigration techniques with the possibilities of preconcentration during extraction from the biological matrix (off-line preconcentration).
- Figures 2 and 3 are of poor quality and need to be redone. In particular, the axis descriptions are not legible.
Answer: Thank you for this comment. We try to improve the quality of Figures.
- Table 2 - What does the acronym DHPG stand for? The structure looks like dihydroxyphenylglycine, but in the text, you talk about 3,4-dihydroxyphenylglycol. In addition, the table does not cite the source from which the pKA values were taken. For the structures reported, I find it very unlikely that any group would be acidic enough to have a negative pKA.
Answer: Thank you for this comment. To determine the properties of the tested analytes, the data contained in web site https://pubchem.ncbi.nlm.nih.gov/
The reviewer correctly noted that the chemical structure given in Table 2 is not 3,4-dihydroxyphenylglycol (DHPG) but dihydroxyphenylglycine. This is of course a mistake. By DHPG we mean the 3,4-dihydroxyphenylglycol. The current version of the table, describing the physicochemical properties of analytes, contains the correct chemical structure of this compound.
- When discussing the choice of pH for LLE, pH values are given to two decimal places. How has pH been measured to this extreme precision? Do you think it is relevant to consider hundredths of a unit of pH if the pH is roughly 3 units different from pKAs?
Answer: We agree with the Reviewer. Although we used the Metler-Toledo pH meter, which gave the measurement with an accuracy of two decimal places, we believe that such accuracy is not really necessary. It was corrected in the manuscript.

Reviewer 3 Report
Comments and Suggestions for Authors
The topic of this article is highly interesting, and the exploration of 3D-printed systems for dispersive solid-phase extraction represents a novel trend. However, the current article lacks significant analytical method development, and in its present state, I do not believe it is suitable for publication. Below are some key points that highlight major issues requiring revision:
1. The authors should try to reduce plagiarism as iThedicator detect 41% similarity index.
2. The authors should avoid long paragraphs. For instance, in the introduction section, the authors can begin a new paragraph starting from line 40 with “The compounds...” for more clarity.
3. Consider removing Figure 1 from the introduction section and instead use it as a graphical figure. Additional experimentation explaining the work done would further enhance its relevance.
4. While the introduction provides substantial emphasis on DHPG, MHPG, and VMA, the extraction methods (SPE, 3D-DSPE, and LLE) are not sufficiently discussed. Comparative studies on these extraction methods should be added from result section to justify their selection over others, with an emphasis on their advantages and disadvantages concerning the compounds under investigation.
5. Although many journals present results before methods and materials, I personally find this structure less effective. Presenting methods before results allows readers to follow the work step by step. Nevertheless, I understand this may vary by journal.
6. Can the authors provide insights into whether artificial urine differs from real urine regarding matrix effects for the detection of the target compounds?
7. In line 404, the authors state that solutions were prepared daily and stored at 4ºC. Did they check the stability of the standard solutions after one month in the freezer or after 8–9 hours at 4ºC?
8. In line 408 and section 3.4, references are needed to support the selected method of analysis. If the MEKC method was developed by the authors, validation is necessary to ensure its reliability.
9. From lines 423 to 428, the authors describe rinsing the column before each run, but there is no mention of potential carryover of the analytes. Was this investigated?
10. Section 3.5.1 do not provide reference explaining the selection of extraction conditions. Were optimization experiments conducted to determine these conditions?
11. In lines 446–447, you state that the SPE columns did not require prior activation according to the manufacturer’s instructions. Please provide a reference for this claim. Additionally, why was DVB activated in DSPE but not in SPE? The comparison between these extraction methods should be consistent.
12. Remove the extra spaces in lines 459 and 462. Begin a new paragraph in line 133 with "An effective...". Add a period to line 146.
13. In section 3.6, the method is stated as not being validated. How can it be claimed as a promising new analytical approach without validation? No recoveries, LODs, LOQs, or calibration curves are provided. A single run without validation is insufficient to substantiate such a claim.
14. The rationale for using 3D-DSPE is unclear. If SPE cartridges with the same material are commercially available, what is the advantage of 3D printing them? If the material is different somehow, this should be clarified.
15. Lines 136–137 should be moved to the introduction and expanded, given the number of references related to the work. Additionally, Table 1 compares prior studies, which would be better suited for a review article. The information from lines 121–180 does not contribute to the results and would be more appropriate in the introduction after revision.
16. The axes of Figure 2 are unclear. Also, can the authors explain the curves at 4.50 and 5.00? Are they related to the analyte?
17. Section 2.2 does not appear to contribute to the results of the study.
18. In Table 4, recovery results are presented after SPE extraction, but there is no mention of calibration curves. Were they performed? This should be stated, including the range. It seems that recoveries were calculated using peak heights, which is not an appropriate method, especially if only one run was performed without error bars or uncertainty measurements. This likely explains the wide recovery range from 300% to 18.2%.
19. Figure 3 is unclear and should be improved for better readability.
Comments on the Quality of English LanguageEnglish language can be improved.
Author Response
Reviewer 3
The topic of this article is highly interesting, and the exploration of 3D-printed systems for dispersive solid-phase extraction represents a novel trend. However, the current article lacks significant analytical method development, and in its present state, I do not believe it is suitable for publication. Below are some key points that highlight major issues requiring revision:
We appreciate the reviewer's interest in the manuscript and their valuable suggestions for improving its quality. In response to these suggestions, we have made efforts to clarify any doubts and enhance the manuscript. It has been completely rewritten, taking into account all of the reviewer's recommendations. We kindly ask that you re-evaluate the revised version.
- The authors should try to reduce plagiarism as iThedicator detect 41% similarity index.
Answer: We sincerely apologize for this inaccuracy. We relied too much on our earlier publications on the subject of catecholamines. We hope that the current version of the work meets the required criteria and does not show similarities.
- The authors should avoid long paragraphs. For instance, in the introduction section, the authors can begin a new paragraph starting from line 40 with “The compounds...” for more clarity.
Answer: Thank you, Reviewer, for your suggestion. We have revised the introduction by removing the overly detailed paragraphs about biological activity of analytes, retaining only the essential information in the relevant sections. In their place, we have significantly expanded the sections focused on the analytical methods used to isolate these molecules from biological matrices. This revision aligns the discussion more closely with the primary objective of our work, which is to consider advantages and disadvantages, optimize and assess the efficiency of selected extraction techniques for the isolation of DHPG, MHPG, and VMA from human urine samples.
- Consider removing Figure 1 from the introduction section and instead use it as a graphical figure. Additional experimentation explaining the work done would further enhance its relevance.
Answer: We agree with the reviewer’s opinion. The figure illustrating the metabolic pathway of the studied analytes could be incorporated into the design of the graphical abstract. In current version of the manuscript we present more data and results documenting the experiments we conducted.
- While the introduction provides substantial emphasis on DHPG, MHPG, and VMA, the extraction methods (SPE, 3D-DSPE, and LLE) are not sufficiently discussed. Comparative studies on these extraction methods should be added from result section to justify their selection over others, with an emphasis on their advantages and disadvantages concerning the compounds under investigation.
Answer: Thank you for this comment. In the revised manuscript, we addressed the advantages and disadvantages of the individual extraction techniques and evaluated the suitability of the procedures used in our study—LLE, SPE, and 3D-DSPE—for the analysis of DHPG, MHPG, and VMA. These evaluations are presented in both the Introduction and the Results and Discussion sections.
- Although many journals present results before methods and materials, I personally find this structure less effective. Presenting methods before results allows readers to follow the work step by step. Nevertheless, I understand this may vary by journal.
Answer: We fully agree with the Reviewer’s suggestion. We believe that presenting the methods before the results allows readers to follow the work more effectively, step by step. However, we have adhered to the journal's guidelines, as specified in the instructions for authors, which recommend placing the Results and Discussion section before the Methods and Materials section.
- Can the authors provide insights into whether artificial urine differs from real urine regarding matrix effects for the detection of the target compounds?
Answer: Thank you for this comment.
Artificial urine and real urine can indeed differ in terms of matrix effects, especially when it comes to the detection of target compounds such as drugs or metabolites. Matrix effects refer to the influence of co-eluting substances from the sample matrix on the analytical detection of target analytes, potentially leading to ion suppression or enhancement in techniques like liquid chromatography-mass spectrometry (LC-MS).
Real urine is a complex biological fluid containing a wide variety of endogenous compounds such as proteins, salts, urea, creatinine, metabolites, and organic compounds whereas artificial urine is typically designed to mimic the main components of real urine (such as urea, creatinine, and electrolytes), but it lacks many of the minor metabolites and proteins found in natural urine. It may also have different pH, ionic strength, and osmolarity compared to real urine samples.
Real urine contains a variety of interfering substances that can affect ionization and analyte detection, in methods like LC-MS/MS. These can cause ion suppression or enhancement, affecting the accuracy and sensitivity of drug detection. Artificial urine is less complex and typically contains fewer interfering substances.
Artificial urine is useful for method development and validation, but it may not perfectly replicate the real matrix effects observed with natural urine, particularly for complex drug metabolites.
Moreover, subtle differences in pH or electrolyte composition between artificial and real urine can also affect the extraction efficiency or stability of target compounds during the detection process.
Artificial urine is often used for method development because it provides a consistent, reproducible matrix that can be used to optimize detection and separation conditions.
However, we are aware that for final method validation, it is important to test with real urine samples to accurately assess matrix effects under actual analytical conditions.
In summary, matrix effects are particularly important when developing a separation method, as interfering substances from the matrix can disrupt signals from the analyte. This applies to most high-performance separation techniques, including electromigration techniques. Matrix effects are less significant during the analyte isolation procedure, as the focus in this case is on finding methods that allow for the most selective isolation of analytes with specific properties. In the case of biological samples, particularly whole blood, serum, or plasma, proteins are usually the main interfering substances in the analysis. Therefore, preliminary steps, such as protein precipitation, are often employed.
The normal pH of urine in a healthy person ranges from 4.5 to 8, with urine typically being slightly acidic. However, the pH can fluctuate due to factors such as diet, medications, or certain medical conditions. Therefore, the influence of this parameter for the method development was considered. It should be noted that the pH value of urine affects the stability of analytes. Literature data shows that catecholamines are particularly sensitive to slight changes in pH, and many sources, including laboratory guidelines, indicate that urine samples should be adjusted to pH 3 to ensure their stability during storage [1]. However, a separate issue is the pH value that promotes optimal isolation of these analytes from the sample matrix, which is related to the pKa value and the logP of individual compounds. For this reason, this parameter was tested in our study.
We are aware that confirming the applicability of the proposed solutions requires full validation and the use of real urine samples. This is the focus of our efforts in the near future. However, conducting such studies in a larger scale requires approval from the relevant Bioethics Committee.
[1] N.B. Roberts, G. Higgins, M. Sargazi, A study on the stability of urinary free catecholamines and free methyl-derivatives at different pH, temperature and time of storage, Clin. Chem. Lab. Med. 48 (2010) 81–87.
- In line 404, the authors state that solutions were prepared daily and stored at 4ºC. Did they check the stability of the standard solutions after one month in the freezer or after 8–9 hours at 4ºC?
Answer: Thank you for this comment. Working solutions with a concentration of 100 µg/mL for all studied analytes were prepared daily from stock solutions with a concentration of 1 mg/mL. Each day, before starting the separation analysis of the samples subjected to the various types of extraction, an analysis of a standard mixture of the studied substances at a concentration of 10 µg/mL was performed. This allowed us to monitor the instrument’s response to the analyte concentration (peak height and area). The repeatability of the results indicated the stability of the analytes over the course of the experiment.
The stability of the instrumental conditions was monitored by observing the current flowing through the capillary under a constant voltage of 25 kV, as well as by observing the repeatability of the analyte migration times in the capillary under constant parameters (internal diameter and length of the capillary, analytical wavelength, applied voltage, temperature of the separation system) and constant composition of the separation buffer.
Changes in the instrument's response to a constant analytes concentration, while the instrumental parameters remained stable, allowed us to assess the stability of the standard solutions. After approximately one month of storing the stock solutions at -20ºC, we observed a decrease in the signal intensity of the analytes by a dozen or so percent. The signal intensity of the analytes in the working solutions at 100 µg/mL, stored at 4ºC, showed a decrease on the following day of analysis. For this reason, the working solutions were prepared daily. However, several months stability studies for analyte solutions were not performed.
- In line 408 and section 3.4, references are needed to support the selected method of analysis. If the MEKC method was developed by the authors, validation is necessary to ensure its reliability.
The buffer composition used in this work was developed by our team several years ago and has been successfully used with minor modifications in our subsequent projects concerning the determination of catecholamines. This separation buffer consisting of 5 mM borax, 150 mM boric acid, 50 mM SDS, and 15% methanol, was previously developed and applicated to separation of seven catecholamines, as described in our paper Miękus et al. [1]. The compounds included adrenaline (A), noradrenaline (NA), dopamine (DA) metabolites: dihydroxyphenylacetic acid (DOPAC), 3-methoxy-4-hydroxyphenyl glycol (MHPG), dihydroxyphenylglycol (DHPG), metanephrine (M), normetanephrine (NM), vanillylmandelic acid (VMA), and homovanillic acid (HVA). In that study, three popular extraction techniques—dispersive liquid-liquid microextraction (DLLME), solid-phase extraction (SPE), and solid-phase microextraction (SPME)—were tested and reported, achieving satisfactory separation of all seven analytes with a capillary length of 60.2 cm and 75 μm i.d.
In the subsequent paper (Kaczmarczyk et al. 2022), we explored the impact of imidazolium-based ionic liquids (ILs) as buffer additives to enhance the separation efficiency of these analytes. Various ILs were individually added to the background electrolyte (BGE), comprising 5 mM borax, 150 mM boric acid, 50 mM SDS, and 15% methanol, at concentrations from 1 to 20 mM. The results indicated that the most effective ILs for improving electrophoretic separation contained chloride anions (1-hexyl-3-methylimidazolium chloride [HMIM+Cl−] and 1-ethyl-3-methylimidazolium chloride [EMIM+Cl−]) and tetrafluoroborate anions (1-hexyl-3-methylimidazolium tetrafluoroborate [HMIM+BF4−]).
In our forthcoming publication [3], we will demonstrate the applicability of these developed separation conditions for the analysis of homovanillic acid (HVA), vanillylmandelic acid (VMA), metanephrine (M), and normetanephrine (NM) in real urine samples from neuroblastoma patients. We confirmed that ionic liquids such as [HMIM+Cl−] and [HMIM+BF4−] show promise as BGE additives and dynamic capillary coating agents for enhanced electrophoretic separation. Validation data indicated strong linearity (R² > 0.996) for all analytes within the concentration range of 0.25–10 µg/mL.
In this current study, we aim to leverage the potential of this separation buffer to assess extraction efficiency and signal amplification for another clinically significant group of catecholamine metabolites. Additionally, due to the smaller number of analytes, we used a shorter capillary length (60.2 cm and 75 μm i.d.), which allowed for shortening the analysis time without losing separation efficiency.
[1] N. Miękus, A. Plenis, M. Rudnicka, N. Kossakowska, I. Olędzka, P. Kowalski, T. Bączek, Extraction and preconcentration of compounds from the l-tyrosine metabolic pathway prior to their micellar electrokinetic chromatography separation, J. Chromatogr. A, 1620 (2020), 461032
[2] N. Kaczmarczyk, J. Ciżewska, N. Treder, N. Miękus, A. Plenis, P. Kowalski, A. Roszkowska, T. Bączek, I. Olędzka, The critical evaluation of the effects of imidazolium-based ionic liquids on the separation efficiency of selected biogenic amines and their metabolites during MEKC analysis, Talanta 238 (2022) 122997
[3] N. Kaczmarczyk, N. Treder, P. Kowalski, A. Plenis, A. Roszkowska, T. Bączek, I. Olędzka, Investigation of Imidazolium-Based Ionic Liquids as Additives for the Separation of Urinary Biogenic Amines via Capillary Electrophoresis, Separations 2023, 10, 116. doi.org/10.3390/separations10020116
- 9. From lines 423 to 428, the authors describe rinsing the column before each run, but there is no mention of potential carryover of the analytes. Was this investigated?
Answer: Electromigration-based procedures require capillary rinsing after each run to ensure proper repeatability of migration times of individual analyses. Any changes can be quickly observed, as they affect the migration times of the analytes or the current flowing through the capillary. In extreme cases of improper capillary rinsing and analyte adsorption onto the capillary wall, significant disruptions in current flow may occur, potentially leading to the interruption of the analysis. In such cases, additional rinsing of the capillary is performed under increased pressure (the standard pressure during rinsing is 20 psi) and for an extended period (the standard rinsing time for one type of solution is 1 to 2 minutes).
Regarding the potential carry-over effect, baseline control was also performed by analyzing the separation buffer alone. To check for any residual analytes carried over from one sample to the next, a blank injection (pure separation buffer) was performed after the analysis of a sample containing high concentrations of the analytes. No analyte signals were detected, indicating that there was no carry-over from the previous sample.
Additionally, carry-over can be prevented by regularly rinsing the capillary with appropriately selected solvents before and after the analysis. Insufficient rinsing may result in analyte residues remaining in the capillary. Whenever carry-over occurs, it is important to consider modifying the working conditions, controlling the filling level of the vials with buffer, rinsing water and regeneration fluid, and such as increasing the rinsing time, using stronger solvents, or optimizing the sample preparation procedures.
- Section 3.5.1 do not provide reference explaining the selection of extraction conditions. Were optimization experiments conducted to determine these conditions?
Answer: This sentence has been corrected.
- In lines 446–447, you state that the SPE columns did not require prior activation according to the manufacturer’s instructions. Please provide a reference for this claim. Additionally, why was DVB activated in DSPE but not in SPE? The comparison between these extraction methods should be consistent.
Answer: Thank you for this comment. Indeed, Evolute® Express ABN, 30mg/1mL; Evolute® Express AX 30mg/1mL; Evolute® Express CX 30mg/1mL were), purchased from Biotage (Uppsala, Sweden and also Styren-DVB (Styre screen), purchased from Supelco/Sigma-Aldrich (St. Louis, USA). were recommended by the manufacturer as not requiring prior activation.
https://www.biotage.com/hubfs/bynder/Document/UI330.V.1-biotage-evolute-express-user-guide.pdf
and recommendation for Styre screen columns from User Guide
- Remove the extra spaces in lines 459 and 462. Begin a new paragraph in line 133 with "An effective...". Add a period to line 146.
Answer: Thank you for this suggestion. The extra spaces in lines 459 and 462 have been removed. The other corrections have been made; however, a significant portion of section 2.1 (Significance of the selection of biological matrices and extraction procedure parameters) has ultimately been deleted from the current version of the manuscript.
- In section 3.6, the method is stated as not being validated. How can it be claimed as a promising new analytical approach without validation? No recoveries, LODs, LOQs, or calibration curves are provided. A single run without validation is insufficient to substantiate such a claim.
Answer: Analyte signal intensity can be used as a preliminary assessment of extraction efficiency in certain cases, particularly during method development and optimization. However, it is not a substitute for a full validation, which ensures the method's accuracy, precision, selectivity, and robustness. Signal intensity alone can provide useful when:
Method Optimization:
- During the early stages of developing an extraction method, analyte signal intensity can help compare different extraction procedures (e.g., solvent choice, pH value, extraction time) to identify the most effective conditions for analyte recovery.
- This can be particularly useful when screening multiple methods to narrow down options before moving into full method validation.
Process Adjustments:
- If minor changes are made to the extraction protocol (e.g., minor variations in reagent concentration or mixing time), signal intensity can indicate whether these adjustments are having a positive or negative impact on analyte recovery.
Qualitative Comparisons:
- When comparing extraction methods qualitatively (e.g., liquid-liquid extraction vs. solid-phase extraction), signal intensity can help assess which method yields higher analyte recovery or cleaner extracts before conducting rigorous validation.
For final conclusions about extraction efficiency, a full validation is needed to assess critical parameters such as accuracy, precision, recovery, matrix effects, and limits of detection/quantitation.
References:
https://www.fda.gov/files/drugs/published/Bioanalytical-Method-Validation-Guidance-for-Industry.pdf
"Principles and Practice of Bioanalysis" (2nd Edition) by Richard F. Venn: Chapters on analytical method validation, which include discussions of the use of the analyte intensity signal during the initial stages of method development through to full validation procedures.
Many research articles compare different extraction techniques, using signal intensity to evaluate preliminary results and suggesting full validation as the next step. Examples:
- Chambers, E., Wagrowski-Diehl, D.M., Lu, Z., & Mazzeo, J.R. (2007). "Systematic and comprehensive strategy for reducing matrix effects in LC/MS/MS analyses." Journal of Chromatography B, 852(1-2), 22-34.
- Matuszewski, B.K., Constanzer, M.L., & Chavez-Eng, C.M. (2003). "Strategies for the assessment of matrix effect in quantitative bioanalytical methods based on HPLC-MS/MS." Analytical Chemistry, 75(13), 3019-3030.
- Souverain, S., Rudaz, S., & Veuthey, J.L. (2004). "Matrix effect in LC-ESI-MS and LC-APCI-MS with off-line and on-line extraction procedures." Journal of Chromatography A, 1058(1-2), 61-66.
These sources provide theoretical and practical justification for using signal intensity in the optimization phase of analytical methods and clearly emphasize the need for full validation before using a given method for analytical purposes.
- The rationale for using 3D-DSPE is unclear. If SPE cartridges with the same material are commercially available, what is the advantage of 3D printing them? If the material is different somehow, this should be clarified.
Answer: Thank you for this suggestion, and we sincerely apologize if our description may have caused any confusion regarding the procedure used. In the current version of the manuscript, we have made an effort to clarify this. In the study, we used a new DSPE extraction kit designed by our team and 3D-printed. In this kit, a commercially available Dow Styrene DVB sorbent was placed and then activated prior to performing the extraction procedure. In the current version of the manuscript, we have also added information regarding the advantages of using of a commercially available sorbent in a 3D printed SPE device.
Moreover, in 3D-DSPE more favorable results for the same type of sorbent and desorbent used in SPE procedure, may be attributed to a looser distribution of sorbent beads in the DSPE procedure, which increases the contact surface area with the sample matrix and the analytes contained within. In our 3D-DSPE procedure, both the extraction and desorption processes were enhanced by mechanically mixing the sample on a magnetic stirrer.
- Lines 136–137 should be moved to the introduction and expanded, given the number of references related to the work. Additionally, Table 1 compares prior studies, which would be better suited for a review article. The information from lines 121–180 does not contribute to the results and would be more appropriate in the introduction after revision.
Answer: Thank you for this comment. According to the reviewer’s suggestions, the corrections have been made. Additionally, we agree that Table 1, comparing previous studies, would be more suitable for a review article, so we have decided to remove it.
- The axes of Figure 2 are unclear. Also, can the authors explain the curves at 4.50 and 5.00? Are they related to the analyte?
Answer: The curvature in the absorbance baseline observed between 4.5 and 5 minutes in the electropherogram corresponds to the electroosmotic flow region. This interference arises due to differences in the composition between the running buffer and the sample buffer.
- Section 2.2 does not appear to contribute to the results of the study.
Answer: This section was removed.
- In Table 4, recovery results are presented after SPE extraction, but there is no mention of calibration curves. Were they performed? This should be stated, including the range. It seems that recoveries were calculated using peak heights, which is not an appropriate method, especially if only one run was performed without error bars or uncertainty measurements. This likely explains the wide recovery range from 300% to 18.2%.
Answer: The table of results has been revised and divided into sections corresponding to each evaluated extraction technique for improved clarity. The results have also been supplemented with standard deviation values, as each approach was repeated at least three times. In the previous version, only the average values were provided. Regarding the potential for analyte signal amplification, enrichment factor (EF) values are reported only when the extraction efficiency exceeded 100%, achieved through both effective isolation conditions and the final dissolution of the sample in just 50 microliters of buffer. This criterion ensured that signal amplification was present. We hope that this revised data presentation is more appropriate. Extraction efficiency values were calculated based on peak heights because in the developed method peak heights were found to be more repeatable than surface area values determined by the apparatus.
- Figure 3 is unclear and should be improved for better readability.
Answer: We decided to remove this figure from the current version of the paper and, in its place, propose a workflow of the 3D-DSPE extraction procedure (as Figure 1).

Round 2
Reviewer 1 Report
Comments and Suggestions for Authors
Please see below
Author Response
Reviewer(s)' Comments to Author:
Reviewer 1
|
Yes |
Can be improved |
Must be improved |
Not applicable |
|
|
Does the introduction provide sufficient background and include all relevant references? |
(x) |
( ) |
( ) |
( ) |
|
Is the research design appropriate? |
(x) |
( ) |
( ) |
( ) |
|
Are the methods adequately described? |
(x) |
( ) |
( ) |
( ) |
|
Are the results clearly presented? |
(x) |
( ) |
( ) |
( ) |
|
Are the conclusions supported by the results? |
(x) |
( ) |
( ) |
( ) |
Comments and Suggestions for Authors
Please see below
Answer: There are no comments attached from the Reviewer.
We would like to thank the Reviewer for evaluating our manuscript. We hope that all the changes introduced have been accepted. We did not find any additional comments from the Reviewer that require clarification.

Reviewer 2 Report
Comments and Suggestions for Authors
The authors have corrected gross errors and rewritten unclearly worded passages. I recommend the article in its present form for publication.
Author Response
Reviewer 2
|
Yes |
Can be improved |
Must be improved |
Not applicable |
|
|
Does the introduction provide sufficient background and include all relevant references? |
( ) |
(x) |
( ) |
( ) |
|
Is the research design appropriate? |
( ) |
(x) |
( ) |
( ) |
|
Are the methods adequately described? |
( ) |
(x) |
( ) |
( ) |
|
Are the results clearly presented? |
( ) |
(x) |
( ) |
( ) |
|
Are the conclusions supported by the results? |
(x) |
( ) |
( ) |
( ) |
Comments and Suggestions for Authors
The authors have corrected gross errors and rewritten unclearly worded passages. I recommend the article in its present form for publication.
Answer: We would like to express our gratitude to the reviewer once again for their insightful and constructive feedback on our manuscript. We have carefully addressed each of the reviewer suggestions, making substantial revisions throughout the manuscript.

Reviewer 3 Report
Comments and Suggestions for Authors
The authors have significantly improved the manuscript, and their efforts are greatly appreciated, as they have effectively addressed all my questions and comments. However, I have noted a few issues that still need to be revised before publication:
1.The similarity index is still detected at 41%, which must be reduced to address plagiarism concerns, even though the authors are citing their own work.
2.In the revised version, when abbreviations are used, the authors must provide the full name the first time they appear. For example: MHPG, DHPG, VMA, HLB, MAX, C18, SPE, LLE, DSPE, etc.
3.I completely agree with the authors' response to the following point: "Can the authors provide insights into whether artificial urine differs from real urine regarding matrix effects for the detection of the target compounds?" I believe their answer and the explanation provided should be incorporated into the Results and Discussion section of the manuscript.
4.Using peak height instead of peak area as a basis for extraction efficiency calculations is uncommon and would require strong justification, as peak area is generally considered a more reliable measure of total analyte concentration. Peak height can fluctuate due to minor changes in detector response or other factors that do not necessarily correlate with analyte concentration. This makes peak area less susceptible to noise and provides a more consistent basis for quantification, particularly in cases where peak shapes are less than ideal. Could the authors explain why the SPE recoveries reached 350% and varied so significantly, even when the same material was used? How those reoveries are better?
Author Response
Reviewer 3
|
Yes |
Can be improved |
Must be improved |
Not applicable |
|
|
Does the introduction provide sufficient background and include all relevant references? |
( ) |
(x) |
( ) |
( ) |
|
Is the research design appropriate? |
( ) |
(x) |
( ) |
( ) |
|
Are the methods adequately described? |
( ) |
(x) |
( ) |
( ) |
|
Are the results clearly presented? |
( ) |
(x) |
( ) |
( ) |
|
Are the conclusions supported by the results? |
(x) |
( ) |
( ) |
( ) |
Comments and Suggestions for Authors
The authors have significantly improved the manuscript, and their efforts are greatly appreciated, as they have effectively addressed all my questions and comments. However, I have noted a few issues that still need to be revised before publication:
1.The similarity index is still detected at 41%, which must be reduced to address plagiarism concerns, even though the authors are citing their own work.
Answer: As the Reviewer has likely noticed, the revised manuscript during the first round of revisions differs significantly from the original. Numerous changes were made in response to the reviewers' suggestions, and all modifications are marked in red. The manuscript text was extensively restructured, and many linguistic corrections were applied. We have made every effort to ensure that the current version is thoroughly revised and substantially restructured.
The Editor's office has informed us: "As for the report submitted by Reviewer, we wish to assure you that we've double-checked the similarity index of the resubmitted manuscript and the result confirms that this issue has been rectified during the first round of revisions".
2.In the revised version, when abbreviations are used, the authors must provide the full name the first time they appear. For example: MHPG, DHPG, VMA, HLB, MAX, C18, SPE, LLE, DSPE, etc.
Answer: Thank you for this comment. We have read the entire manuscript very carefully once again and have supplemented the abbreviations by providing their full names where they first appeared.
3.I completely agree with the authors' response to the following point: "Can the authors provide insights into whether artificial urine differs from real urine regarding matrix effects for the detection of the target compounds?" I believe their answer and the explanation provided should be incorporated into the Results and Discussion section of the manuscript.
Answer: Thank you for this comment. According to Reviewer suggestion the explanation concerning matrix effects for the detection of the target compounds were introduced into the current version of the manuscript (section 2.1).
4.Using peak height instead of peak area as a basis for extraction efficiency calculations is uncommon and would require strong justification, as peak area is generally considered a more reliable measure of total analyte concentration. Peak height can fluctuate due to minor changes in detector response or other factors that do not necessarily correlate with analyte concentration. This makes peak area less susceptible to noise and provides a more consistent basis for quantification, particularly in cases where peak shapes are less than ideal. Could the authors explain why the SPE recoveries reached 350% and varied so significantly, even when the same material was used? How those reoveries are better?
Answer: To answer the first part of the question. In capillary electrophoresis (CE), peak height can be used as an alternative to peak area for analyte quantification. Although peak area is generally preferred because it is less affected by variations in peak shape and width, peak height may offer certain advantages :
- In cases where peaks are sharp and narrow, peak height can be more directly proportional to analyte concentration, providing reliable quantification without the need for longer analysis times that are sometimes necessary for stable peak areas.
- For analytes present at very low concentrations, peak height might offer improved sensitivity over peak area. This is because in CE, small, well-defined peaks with minimal baseline noise may lead to greater signal-to-noise ratios using peak height.
- When minor fluctuations in migration times occur (for example, due to temperature shifts or slight variations in buffer composition), peak height is sometimes less impacted than peak area, especially when these fluctuations lead to slight peak broadening.
While peak area remains the standard for most CE applications, peak height can be useful as an alternative, particularly when peak shapes are consistent, and sensitivity is a priority.
In the developed electrophoretic method for the determination of selected NE metabolites, we obtained slightly better RSD (%) values for height values compared to the surface areas of their corresponding peaks (Table below). Therefore, the peak height values of analytes were selected for further calculations
Tab. Normalized peak height and peak area RSD (%) values for analytes of interest.
|
Analytes |
RSD values (%) (n=6) |
||
|
Migration time |
Peak area |
Peak height |
|
|
DHPG |
1.3 |
7.1 |
6.7 |
|
MHPG |
1.4 |
8.8 |
7.9 |
|
VMA |
1.2 |
7.9 |
7.4 |
To answer the second part of the question. In some extraction techniques used for isolating analytes from biological matrices, recoveries exceed 100% when preconcentration techniques are employed. The recovery values presented reflect the degree of off-line preconcentration related to the isolation of analytes from the urine matrix.
To explain further: during extraction, certain methods enable the isolation and concentration of analytes from a larger sample volume into a smaller final volume. This results in an increased analyte concentration in the extract, which can sometimes be interpreted as a recovery greater than 100%. In such cases, the recovery calculation reflects both the efficiency of analyte isolation and the concentrating effect of the extraction technique, rather than the actual proportion of analyte originally present in the sample. Indeed, for 3D-DSPE higher values of standard deviation were noted for higher extraction efficiency values. This may probably be due to the fact that the commercially available Dow Styrene DVB sorbent was weighed and placed in the printed system. Even small differences in the sorbent weight may translate into increased SD values of extraction efficiency. However, the general trend shows a great potential of the 3D-DSPE technique in the extraction process of DHPG, MHPG and VMA, giving at the same time the greatest possibilities of increasing the signal intensity.
